# PERSonalised Incentives for Supporting Tobacco cessation (PERSIST) among healthcare employees: a randomised controlled trial protocol

Nienke W Boderie [ID],[1] Johannes LW van Kippersluis,[2,3] Diarmaid T Ó Ceallaigh,[2] Márta K Radó [ID],[1,4] Alex Burdorf [ID],[1] Frank J van Lenthe,[1] Jasper V Been[1,4]

¹Department of Public Health, Erasmus MC, University Medical Centre Rotterdam, Rotterdam, The Netherlands
²Erasmus School of Economics, Erasmus University Rotterdam, Rotterdam, The Netherlands
³Tinbergen Institute, Amsterdam, The Netherlands
⁴Department of Paediatrics, Division of Neonatology, Erasmus MC – Sophia Children's Hospital, University Medical Centre Rotterdam, Rotterdam, The Netherlands

**Correspondence to**
Dr Jasper V Been;
j.been@erasmusmc.nl

## ABSTRACT

**Background** Smoking is the primary preventable risk factor for disease and premature mortality. It is highly addictive and cessation attempts are often unsuccessful. Incentive-based programmes may be an effective method to reach sustained abstinence. Individualisation of incentives based on personal characteristics yields potential to further increase the effectiveness of incentive-based programmes.

**Method** A randomised controlled trial among healthcare workers recruited through their employer and signed up for a group-based smoking cessation programme. The intervention under study is the provision of personalised incentives on validated smoking cessation at several time points after the smoking cessation programme. A total of 220 participants are required. Participants are randomised 1:1 into intervention (personalised incentives) or control (no incentives). All participants join the group-based programme. Incentives are provided on validated abstinence directly after the smoking cessation programme and after 3, 6 and 12 months. Incentives are provided according to four schemes: (1) Standard: total reward size €350, pay-out scheme: €50 (t=0), €50 (t=3 months), €50 (t=6 months) and €200 (t=12 months), (2) descending: total reward size €300, pay-out scheme: €150, €100, €50 and €0, (3) ascending: total reward size: €400, pay-out scheme: €0, €0, €50 and €350 and (4) deposit: total reward size €450, pay-out scheme: €50, €50, €150, €200; participants pay a €100 deposit, returned conditional on abstinence after 6 months.

Advice on which incentive scheme suits participants best is based on willingness to provide a deposit, readiness to quit, nicotine dependency and long-term or short-term reward preference. Participants are free to deviate from this advice. Abstinence is validated at each time point, with 15 months of total follow-up. The primary end point is validated abstinence at 12 months. Effectiveness will be determined by intention-to-treat analysis.

**Ethics and dissemination** The Erasmus MC Medical Ethics Committee decided that according to the Dutch Human Research Law (WMO), the protocol required no formal ethical approval. The results will be published in a peer-reviewed scientific journal and communicated to the participants.

### Strengths and limitations of this study

► The PERSonalised Incentives for Supporting Tobacco cessation (PERSIST) trial has a unique hybrid design with a nudged choice embedded within a randomised trial and is the first of its kind to provide a personalised advice on the mode of providing incentives, which likely best fits individual participant characteristics.

► This trial furthermore addresses a knowledge gap regarding long-term effectiveness of incentive-based strategies beyond the actual intervention period.

► A focus on helping hospital employees quit smoking is essential given their exemplary role in promoting health.

► For logistical reasons, participants from the intervention and control arms may attend the same group sessions, potentially leading to spill-over effects.

► Due to the predicted sample size, it is not possible to include an additional intervention arm where incentives are provided but not personalised.

**Trial registration number** Netherlands Trial Register NL7711.

## BACKGROUND

Smoking is the primary preventable risk factor for disease and premature mortality.[1] Smoking is highly addictive and the vast majority of cessation attempts is unsuccessful.[2] In 2017, 38% of smokers in the Netherlands attempted to quit for at least 24 hours.[3] Several methods exist to increase the odds of quitting, such as nicotine replacement therapy (NRT),[4] individual behavioural counselling[5] and group-based smoking cessation programmes.[6] However, despite the relative effectiveness of these approaches, in absolute numbers, the proportion of smokers who quit is still low.

A promising method to encourage smoking cessation is the provision of financial

incentives upon validated abstinence. A recent Cochrane review found high-certainty evidence for increased smoking cessation rates at 6 or 12 months follow-up (risk ratio (RR)=1.49, 95% CI 1.28 to 1.73).[7] Financial incentives make the desired behaviour—in this case smoking cessation—more attractive by rewarding it. Importantly, financial incentives provide a short-term reward for a behavioural change that would normally only provide rewards in the long term. Long-term rewards are often valued less compared with short-term rewards due to declining perceived values of a delayed reward, so-called delay discounting.[8] Typically, people give disproportionally stronger relative weights to instantaneous rewards, so-called present bias,[9] which gives a strong theoretical argument for providing short-term incentives for smoking cessation.

Incentive-based smoking cessation programmes have previously been implemented at the workplace.[10–12] Recently, the 'Continuous Abstinence Through Corporate Healthcare' (CATCH) trial was carried out among 61 Dutch companies, randomising companies to group-based smoking cessation training for smoking employees with or without financial incentives to promote sustained cessation. Among those eligible to receive incentives (up to €350), the proportion of abstinent participants at 1-year follow-up was significantly higher compared with those assigned to group-based training alone (41% vs 26%).[13]

The effectiveness of incentives in promoting sustained smoking cessation depends on how and when rewards are provided. Traditionally, incentives have been tested according to a one-size-fits-all approach, using incentive size as a measure of anticipated effectiveness. A recent US trial comparing four different incentive programmes, however, showed that effectiveness is variable among different types and schemes of incentives.[10] A deposit-based scheme, where participants commit their own money and return is conditional on validated abstinence, was unpopular but resulted in a significantly higher proportion of abstinent individuals compared with the reward-based incentive arm.[10] Further analysis revealed associations between personal characteristics and acceptability and efficacy of incentive programmes.[14] Both readiness-to-quit-smoking and low tendency to discount rewards were associated with increased effectiveness of both reward-based and deposit-based programmes. This suggests that personalisation of incentive programmes yields the potential to further increase the effectiveness of incentives to promote long-term abstinence.

Here we present the study protocol for the 'Personalised Incentives for Supporting Tobacco Cessation' (PERSIST) trial. The trial will be conducted among healthcare employees, for whom smoking cessation is particularly relevant as they may be expected to have an exemplary role. Moreover, in an attempt to work towards a smoke-free generation by 2040, the Dutch National Prevention Agreement requires university hospitals to be smoke free by mid-2020 and all healthcare facilities by 2025.

The aim of this study is to investigate whether personalised incentives in addition to a group-based smoking cessation programme will increase sustained smoking cessation among participants. This unique randomised controlled trial attempts to optimise the effectiveness of incentive programmes by providing personalised advice while leaving participants in full control to choose the incentive scheme of their choice. Our approach will further provide insight into which incentive scheme participants prefer and whether they can be nudged into the programme that most likely fits their personal characteristics best.

## METHODS AND ANALYSIS
### Objectives
The main objective is to evaluate the effectiveness of personalised incentives in combination with group-based training sessions provided in the work environment on sustained smoking abstinence among healthcare workers compared with group-based training sessions alone.

### Methods
The protocol follows the 'Standard Protocol Items: Recommendations for Interventional Trials' (SPIRIT) guidelines.[15]

### Trial design
The PERSIST trial is a randomised controlled non-blinded trial with two parallel arms.

### Study setting
The trial will be implemented at Dutch healthcare institutions. The trial is primarily implemented at the Erasmus MC and is further being extended to Franciscus Gasthuis & Vlietland and Ikazia Hospital. Inclusion of other healthcare institutions in the greater Rotterdam area is ongoing.

### Trial population and eligibility criteria
Employees are eligible if they are (1) aged 18 years or more, (2) employed at one of the participating healthcare institutions and (3) a daily smoker (ie, smoking at least one cigarette per day). Exclusion criteria are (1) only using e-cigarettes and (2) not being able to physically attend the group-based training sessions.

Participants are recruited through their employer. Information about the training and the PERSIST trial is spread through intranet, email, flyers and screensavers. In collaboration with the organisation responsible for the training (SineFuma, see the Interventions section), information sessions about the training and trial are organised.

### Interventions
Eligible participants are individually randomised 1:1 into a control arm, receiving no incentives, and an intervention arm, receiving personalised incentives. See figure 1 for a flowchart. Both arms participate in a group-based smoking cessation training provided

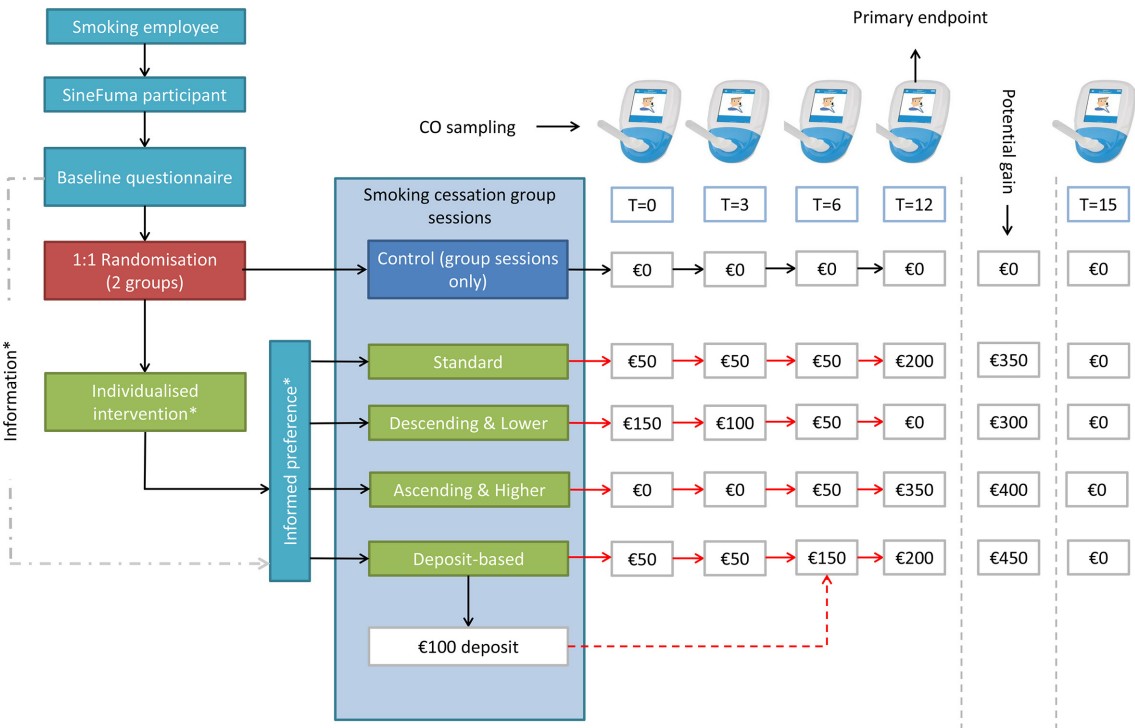

**Figure 1** Flow diagram of participant inclusion plus outline of control and intervention conditions. *Participants are provided with an informed choice regarding the individualised incentive scheme based on: 1. Degree of tobacco dependence, 2. Readiness to quit and 3. Present-bias. Note: red arrows are conditional on sustained biochemically validated smoking cessation.

by SineFuma, a company specialised in supporting smoking cessation (www.rookvrijookjij.nl). Groups consist of 8–16 participants and the training consists of seven sessions of approximately 90 min duration held over a period of 8 weeks. Participants in the intervention and control arms participate in the same training sessions for logistical reasons. Group-based training sessions will start as soon as at least eight people have signed up to avoid participants lose motivation due to long waiting times.

Group sessions are provided at locations within each of the participating healthcare institutions. The aim of the training is to quit smoking as a group: participants jointly quit smoking in the third session. The follow-on sessions then focus on supporting sustained cessation through the first few weeks following the quit attempt. The participants and coach discuss various types of NRT as part of the training. No NRT is provided in the trial but participants are free to use an NRT of their choice. During each session, exhaled carbon monoxide (CO) measurements (see the Outcomes section for more information) are used as a motivator to stay abstinent. The SineFuma group-based training is recognised as an efficient lifestyle intervention by the Dutch Government,[16] and participation costs are reimbursed by basic insurance. As such, the control arm can be considered as usual care. If a portion of the costs are at the employee's own expense through the deductible, employers have agreed to pay for the training.

## Incentives

The difference between intervention and control conditions lies within the provision of incentives. Participants in the intervention arm who remain continuously abstinent following the joint quit attempt are eligible to receive incentives according to a fixed time schedule. Incentives are provided as vouchers representing monetary value. Vouchers can be spent at online web shops or on events such as a high tea in a local café or a night at a hotel and so on. Vouchers cannot be spent on tobacco products. The first incentive is provided directly after the group-based smoking cessation training, marking the moment from which participants have to remain abstinent independently.

In order to personalise incentives, four different incentive schemes are offered, which participants are free to choose from. Across the different schemes, the monetary value of the incentives varies over time, and the potential total financial reward differs. The schemes are as follows (see figure 1):

1. The standard scheme. Analogous to the CATCH trial,[13] vouchers with a monetary value of €50, €50, €50 and €200 are offered on validated abstinence at t=0 month (ie, directly following the final smoking cessation training session), t=3 months, t=6 months and t=12 months, respectively. Hence, the maximum total amount of incentives that can be earned is €350.

2. The descending scheme. Offer vouchers with a monetary value of €150, €100 and €50 on validated ab-

stinence at t=0 month, t=3 months and t=6 months, respectively. No further incentives are provided at t=12 months. The maximum total amount of incentives is €300.

3. The ascending scheme. Offer vouchers with a monetary value of €50 and €350 on validated abstinence t=6 months and t=12 months, respectively. No incentives are provided at t=0 and t=3 months. The maximum total amount of incentives is €400.

4. The deposit-based scheme. Participants are asked to provide a €100 deposit at the beginning of the trial. The deposit scheme offers vouchers with a monetary value of €50 on validated abstinence at both t=0 and t=3 months. On validated abstinence at t=6 months, participants in this scheme will receive vouchers representing a monetary value of €150; in addition, their €100 deposit will be refunded. On validated abstinence, at t=12 months they will receive vouchers representing a monetary value of €200. Accordingly, the maximum total amount of incentives is €450, excluding the deposit refund. If one does not stay abstinent until 6 months after the last training session, the deposit will be donated to the Netherlands Lung Foundation.

In choosing the monetary amounts of incentives, we followed the general principle that the incentives should be large enough to motivate smoking cessation, but not too large either to avoid crowding out intrinsic motivation and to make scaling-up the trial prohibitively costly.[17] The standard scheme is identical to the scheme in the CATCH trial.[18] Given time discounting due to inflation and individual impatience, it seems natural to apply some compensation in the total potential gains to the descending and ascending schemes. For example, why would someone self-select into an ascending scheme with zero pay-out in the first 6 months if the total potential gain is the same as the descending scheme where one can cash-in €150—already after at baseline? Further, as deposit contracts have proven to be most effective among those who choose one,[10] we aimed to make this scheme the most attractive by providing the largest total reward size.

The key aspect of the intervention is the personalisation of the incentives. The varying values over time and varying total reward sizes allow for differences in reward preferences. For example, those who prefer to earn money sooner can choose accordingly (descending scheme), however, in that case the maximum total amount rewarded will be lower, while patience is rewarded with a higher total amount (ascending scheme). Halpern *et al*[10] showed that in the context of smoking cessation promotion at the workplace, deposit schemes are unpopular but very effective; therefore, the current deposit scheme is designed to be the most attractive by having the largest potential total reward.

In an attempt to maximise the effectiveness of incentive provision, participants receive automated advice regarding which scheme is most likely to fit their personal characteristics. This advice will be based on the following characteristics:

1. Degree of tobacco dependence. Based on the Fagerström Test for Nicotine Dependence.[19] Fagerström score below 5=no/low nicotine dependence or Fagerström score of 5 or higher=nicotine dependent.

2. Readiness-to-quit. Based on Prochaska Stage of Change.[20] Preparation stage (readiness to quit within the next 30 days) or (pre)contemplation stage (ready to quit within the next 6 months or not ready to quit at all).[10 20]

3. Present bias. Measured by temporal discounting magnitude based on the Kirby Scale.[21] Temporal discounting refers to the phenomenon where future rewards are perceived as being worth less than an immediate reward.[21] Based on a 27-item monetary choice questionnaire, those choosing more than 50% of the times the larger delayed rewards are classified as having delayed reward preference and those choosing less than 50% of the times the larger delayed rewards are classified as having present reward preference.[21 22]

4. Willingness to pay a deposit, yes or no.

The first three factors have been associated with the effectiveness of smoking cessation interventions and with reaching sustained abstinence in reward-based programmes.[14] The fourth factor is used as an indicator of acceptability of the deposit-based scheme, as deposit schemes are expected to be not only the most effective but also the least popular. The four factors are dichotomised into 'positive' or 'negative' values in order to advise participants. Positive values are associated with higher quit percentages, that is, being in a preparation stage, having low nicotine dependence and having a delayed reward preference. Negative values are associated with lower quit percentages, that is, being in a (pre)contemplation stage, being nicotine dependent and having an immediate reward preference.

A combination of three positive values leads to the ascending scheme, two positive values to the standard scheme and two or three negative values to the descending scheme (see table 1). Those willing to pay a deposit are automatically advised the deposit-based scheme. The rationale behind this is that those with the lowest scores are likely to experience the most trouble quitting, and hence arguably benefit most from higher rewards in the beginning. Those with a higher score might rather benefit from incentives as a motivator to remain abstinent, and thus receive higher rewards towards the end of the study period. Importantly, participants are, however, free to choose the scheme of their preference when entering the trial regardless of the advice they received.

As participants have freedom of choice over the incentive schemes, this is likely to result in a skewed distribution of participants over the schemes. However, this is not an issue in the current trial as the aim is to estimate the overall effect of personalised incentives compared with no incentives and not to estimate the discrete effects of the

**Table 1** Answer combinations and advised schemes

| Degree of tobacco dependence* | Readiness-to-quit† | Present bias‡ | Willing to pay a deposit | Scheme advice |
|---|---|---|---|---|
| <5 | Ready to quit within the next 30 days | Delayed reward preference | No | Ascending |
| <5 | Ready to quit within the next 30 days | Present reward preference | No | Standard |
| <5 | Ready to quit in the next 6 months or not ready to quit | Delayed reward preference | No | Standard |
| <5 | Ready to quit in the next 6 months or not ready to quit | Present reward preference | No | Descending |
| ≥5 | Ready to quit within the next 30 days | Delayed reward preference | No | Standard |
| ≥5 | Ready to quit within the next 30 days | Present reward preference | No | Descending |
| ≥5 | Ready to quit in the next 6 months or not ready to quit | Delayed reward preference | No | Descending |
| ≥5 | Ready to quit in the next 6 months or not ready to quit | Present reward preference | No | Descending |
| Any answer | Any answer | Any answer | Yes | Deposit |

*Assessed via Fagerström score.
†Assessed via Prochaska state of change.
‡Assessed via Kirby discontinuity score.

individual schemes. In fact, the current trial will provide information as to which schemes are most popular, how this relates to personal characteristics and whether participants may be nudged into the scheme that is assumed to best fit their personal characteristics.

## Outcomes

The primary outcome is CO-validated 12-month continuous abstinence from smoking (Russel's Standard, RS12).[23] The primary outcome is measured at t=12 months after the last group-based training. However, participants jointly quit smoking at the third training session which is approximately 1 month prior to the last training session. This 1-month gap between the joint smoking attempt and the start of our measurement allows participants to have a setback in smoking cessation while they are still in the training. From the end of the training (t=0 month) onwards, all participants are expected to be smoke free. Abstinence is assessed via self-reported abstinence and biochemically validated using expired air (CO) measurements. Self-reported abstinence assesses both point and continuous abstinence. Point prevalence is assessed by 7-day abstinence at each follow-up point and prolonged abstinence via the total continuous abstinence period.[23] Self-reported abstinence is validated via assessment of expired air CO concentrations using a handheld monitor (PiCO Smokerlyzer, Bedfont Scientific, Kent, England). An exhaled-air CO reading of <10 ppm is considered indicative of abstinence.[23] CO quantification is a reliable and non-invasive method to ascertain recent smoking and the preferred method in populations using NRT.[23] In case of disagreement between self-reported abstinence and CO reading, abstinence is defined by the latter.

Secondary outcomes are smoking abstinence directly after the group-based smoking cessation training (t=0), and after 3 months (t=3), after 6 months (t=6) and after 15 months (t=15). The same methods and definitions to determine continuous abstinence are used for all outcomes.

Tertiary outcomes are the incremental cost-effectiveness ratio and the incremental cost-utility ratio, where the first is calculated using the total costs per quitter and the latter by the costs and self-reported quality of life (EurQol Five Dimensions).

## Sample size

The CATCH trial, which employed similar incentive schemes as the standard arm of the current trial, reported 41.1% abstinence at 12-month follow-up in the intervention arm versus 26.4% among participants who only attended the training sessions.[13] The PERSIST trial will be embedded in healthcare institutions, many of which are currently implementing smoke-free policies, which is expected to be an extra external motivator for cessation. As such the validated abstinence percentages are expected to be slightly higher than in the CATCH trial, that is, 45% and 30%, respectively. We expect that personalisation of the incentives will further increase effectiveness to approximately 50%. In order to detect this difference in validated sustained abstinence (ie, 50% vs 30%) with a power of 0.8 and an alpha of 0.05 and using 1:1 randomisation, a total of at least 186 participants are

needed. Taking into account a 15% loss to follow-up due to unexpected employee turnover,[11 24] 220 participants are required. Further taking into account the potential maximum deviation from a 1:1 allocation ratio which may arise due to the stratified block randomisation procedure used, the minimum sample size required will increase in proportion to the number of different hospitals included in the trial. As the trial aims to estimate the effect of personalised incentives compared with no incentives and not to estimate the effect of each individual scheme, skewed allocation among the schemes is not an issue and is therefore not accounted for in the required sample size.

### Recruitment

Recruitment takes place at participating healthcare institutions, starting with Erasmus MC. Potential participants are informed about the group-based smoking cessation training sessions through internal websites, flyers, information sessions provided by SineFuma or by referral through their company doctor. Potential participants receive a letter explaining the different incentive schemes and the research procedures. On enrolment, informed consent is required.

### Allocation

Within participating institutions, participants will be individually randomised 1:1 to either the control or intervention arm. Permuted block randomisation with varying block sizes of four and six is used. There are two stratification factors: participating institution and readiness-to-quit (preparation or contemplation stage). A computer-generated allocation sequence was provided by ALEA randomisation service, in collaboration with the Erasmus MC Clinical Trial Centre. The block sizes will not be disclosed so as to ensure allocation concealment.

ALEA will not release the randomisation result until the patient has been recruited into the trial and all baseline questionnaires have been completed. Participants are informed about their allocation by email. Inherent to the intervention and study design, participants and outcome assessors will not be blinded to allocation. Group session trainers will be blinded to allocation and data will be anonymised before analysis.

### Collection methods

As described in the Outcomes section, self-reported abstinence will be validated using expired air CO concentration readings obtained via a handheld monitor. Participants will be invited to complete web-based questionnaires assessing self-reported quality of life (EurQol Five Dimensions Health Questionnaire),[25] smoking abstinence self-efficacy (Smoking Abstinence Self-efficacy Questionnaire),[26] perceived stress (Perceived Stress Scale)[27] and self-reported smoking abstinence (point prevalence and continuous abstinence)[23] at each time point during follow-up. At baseline, the following items will be collected: educational level following the International Standard Classification of Education (ISCED;[28] 0 (early childhood education) to 8 (doctoral or equivalent level)), individual gross monthly income (<€1500, €1500–1900, €1900–2600, >€2600), age, body mass index, gender and smoking behaviour (pack years, history of quit attempts and smoking history). In order to provide a personalised advise, the following questionnaires are also taken at baseline: nicotine dependency (Fagerström Scale),[19] readiness-to-quit (Prochaska Stages of Change)[20] and present bias (Kirby Discontinuity Scale).[21 29] See table 2 for a schematic overview of data collection during the trial.

Participants will be sent reminders after 5 days for uncompleted questionnaires. A follow-up reminder will be sent 2

| Table 2 | Schematic overview of data collection during the trial | | | | | |
|---|---|---|---|---|---|---|
| **Questionnaire** | **Baseline** | **t=0 month** | **t=3 months** | **t=6 months** | **t=12 months*** | **t=15 months** |
| Demographic variables | X | | | | | |
| Fagerström Scale | X | | | | | |
| Prochaska Stage of Change | X | | | | | |
| Kirby Discontinuity Scale | X | | | | | |
| Willingness to pay a deposit | X | | | | | |
| EQ-5DF† | X | X | X | X | X | X |
| SASEQ‡ | X | X | X | X | X | X |
| Perceived Stress Scale | X | X | X | X | X | X |
| Self-reported abstinence§ | X | X | X | X | X | X |
| Evaluation training sessions | | X | | | | |
| Evaluation PERSIST | | | | | | X |

*t = 12 months is the primary endpoint.
†EurQol Five Dimensions Health Questionnaire.
‡Smoking Abstinence Self-efficacy Questionnaire.
§The self-reported abstinence questionnaire is adjusted to each corresponding time moment, for example, at t =3 the maximum period not smoked is 3 months and so on.
PERSIST, Personalised Incentives for Supporting Tobacco Cessation.

days later. If there are still open items, participants will be contacted by telephone 2 days after the last reminder and a follow-up phone call will be made another 2 days later.

### Data management

Informed consent will be provided digitally through Lime-Survey (LimeSurvey GmbH, Hamburg, Germany). All trial data will be entered electronically using GemsTracker (GemsTracker, Erasmus MC and *Equipe Zorgbedrijven*, Rotterdam, The Netherlands). Data are handled confidentially and stored anonymously, where data need to be linked to an individual subject (eg, linking longitudinal data), a subject identification code will be used. The code will not be based on patient initials and date of birth to preserve anonymity. The key to the code will be safeguarded by the investigator and kept separate from data files. All of the study essential documents will be retained and archived for 15 years after completion of the study. They will be stored securely and adequately protected from damage.

### Statistical methods

The primary analysis compares continuous smoking abstinence rates at 12 months after the group-based training session in the intervention arm against the control arm, following an intention-to-treat analysis. In secondary analyses, continuous smoking abstinence at 15 months is assessed to see if abstinence is sustained after the incentives stopped. All randomised participants will be included in the final analysis, those lost to follow-up will be classified as smoking except for an unavoidable loss to follow-up (eg, untraceable contact information or death, following the RS12).[23] Generalised linear models with a logit link function will be used to assess the difference in the primary outcome between control and intervention arms at t=12 months and to account for clustering between hospitals.

Additionally, subgroup analyses will be performed to identify possible subgroup-specific effects of the intervention. Planned subgroup analyses are based on:
1. Readiness-to-quit. Preparation stage versus contemplation stage.[10 20]
2. Degree of tobacco dependence. Fagerström score below 5 versus Fagerström score of 5 or higher.[19]
3. Present bias. Delayed versus present reward preference. Delayed reward preference=more than 50% larger delayed reward is chosen or present reward preference=less than 50% larger delayed reward is chosen.[21 22]
4. Income. Based on gross monthly income in €. Low income=below 1500 versus moderate/high income=above 1500.[30]
5. Educational level. Based on the ISCED.[28] Lower educated=ISCED 0–4 versus higher educated=ISCED 5–8.

Subgroup analysis is not included in the power analysis and therefore exploratory. For missing data on subgroup information, multiple imputations will be performed.

### Patient and public involvement

Participants were not involved in designing the trial. Once the results are published, participants will be informed about the publication by email. This email will be drafted suitable for a non-specialist audience. Public involvement will be sought through the Erasmus Initiative Smarter Choices for Better Health initiative.

## ETHICS AND DISSEMINATION

The study protocol has been assessed by the Erasmus University Medical Centre Medical Ethics Committee (MEC-2019-0140). The committee decided that according to the Dutch Human Research Law (WMO), the protocol required no formal ethical approval. All participants will provide informed consent for participating in the study.

The results will be published in a peer-reviewed scientific journal. On completion of the trial and after the publication of the results, data requests can be submitted to the corresponding author at Erasmus MC, Rotterdam, The Netherlands.

## DISCUSSION

We present the protocol of the PERSIST trial. The aim of the trial is to evaluate the effectiveness of personalised incentives in addition to group-based smoking cessation training on sustained smoking abstinence among hospital employees compared with group-based training alone. Personalisation entails that participants are advised and can choose from four different incentive schemes: standard, ascending, descending and deposit-based.

### Strengths and limitations

The PERSIST trial is unique in its approach to personalising incentives to promote smoking cessation. Strong links between personal characteristics and effectiveness of incentives to promote smoking cessation have been demonstrated previously, however, to our knowledge, the effectiveness of personalised incentives has not been studied before.[14] Another unique element of the PERSIST trial is that participants receive personalised advice for an incentive scheme, based on personal characteristics, yet are free to deviate from the advised scheme. Voluntary choice with a personalised default is a common tool in behavioural economics to nudge people towards the scheme that is plausibly in their best interest.[31] This feature is scientifically innovative in the context of smoking cessation programmes and importantly lets the treated individuals choose the scheme they will be enrolled.

By focusing on supporting smoking cessation among hospital employees, the PERSIST trial is executed in a highly relevant setting, as in light of the National Prevention Agreement, all university hospitals have to be smoke free by 2020 and all general hospitals by 2025.[32] Finally, the PERSIST trial progresses from previous studies by extending follow-up beyond the last time point when incentives are provided to investigate if participants remain abstinent when incentives are no longer provided.

A limitation of the PERSIST trial is that participants from the intervention and control arm attend the same cessation support training sessions due to practical reasons, as slow

inclusion rates otherwise would have caused long waiting periods between signing up and starting the training. Participants in the control arm might perceive this negatively and potentially drop out, causing attrition bias. Another consequence could be spill-over effects[33]; sharing of vouchers between control and intervention arm, leading to a double underestimation of treatment effectiveness. To avoid this, participants are informed about their status as early as possible to make sure they have time to accept their status. A third arm consisting of non-personalised incentives could have provided additional potentially valuable information on the effectiveness of personalised incentives versus incentives provided according to a fixed scheme. However, this option would have required an unfeasibly large sample size.

**Contributors** JVB, FJVL and JLWvK wrote the initial version of the programme grant request on which this protocol was based. NWB drafted the manuscript and runs the day-to-day study. DOC did the sample size calculations, advised on the randomisation method and drafted the allocation algorithm and MR advised on the statistical analysis. The paper was revised and edited by NWB, JLWvK, MKR, DOC, AB, FJVL and JVB and all authors approved the final manuscript.

**Funding** This study is funded by the Erasmus Initiative Smarter Choices for Better Health and the Erasmus Trustfonds. JVB is funded by a personal fellowship from the Netherlands Lung Foundation.

**Competing interests** None declared.

**Patient and public involvement** Patients and/or the public were not involved in the design, or conduct, or reporting, or dissemination plans of this research.

**Patient consent for publication** Not required.

**Provenance and peer review** Not commissioned; externally peer reviewed.

**ORCID iDs**
Nienke W Boderie http://orcid.org/0000-0002-1600-380X
Márta K Radó http://orcid.org/0000-0002-1676-5951
Alex Burdorf http://orcid.org/0000-0003-3129-2862

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
