## [Reviewer comments · BMJ Open]

ARTICLE DETAILS

TITLE (PROVISIONAL)	Personalised incentives for supporting tobacco cessation among health care employees (PERSIST): a randomised controlled trial protocol
AUTHORS	Boderie, Nienke; van Kippersluis, Hans; Ó Ceallaigh, Diarmaid; Rado, Márta; Burdorf, Alex; Van Lenthe, Frank; Been, JV

VERSION 1 – REVIEW

REVIEWER	Hayden McRobbie The University of New South Wales, Australia
REVIEW RETURNED	01-Mar-2020

GENERAL COMMENTS	This protocol describes a study that aims to examine the effect of personalised financial incentives on smoking cessation outcomes, compared to no incentives. The study aims to provide four different incentive schemes and provides a recommendation on which scheme would be most appropriate depending on a range of different factors. Participants are however free to ignore the recommendation and chose a scheme that they like best. This is a pragmatic approach that lends itself better to implementation in the real world. Overall this is a well-written protocol paper for a study that is presumably already underway. I have a few comments that might be helpful in clarifying a few points. I have listed these in the order in which they appear in the paper. Page 7, line 36: e-cigarettes are not smoked. I suggest stating 'only using e-cigarettes' instead. Page 7: It would be helpful to provide a little more information on the group-based treatment programme. It sounds like withdrawal-oriented treatment. Adding some information about the treatment approach, including the use of smoking cessation medication would be useful. I also note that participants stop smoking on the 3rd session, but the that T0 starts at the end of the group programme, which means that some participants would have achieved 4-weeks of abstinence by this time point. It could be helpful to make this clear to readers, as T12, for example, is actually 13 months of abstinence, not 12 months. It might also be useful to have some discussion on the timing of the incentives in relation to the quit date in the group-based treatment programme. Some will have already progressed through four weeks of abstinence, which is
---

when tobacco withdrawal symptoms are most severe. One might argue that incentivising people earlier in the quit attempt could be useful.

Page 9, line 8: I am slightly perplexed by the inclusion of readiness-to-quit as a personalising factor when the treatment programme appears to require people to be ready to quit. I suppose the offer of financial incentives may quickly change someone's readiness to quit. I notice that the inclusion/exclusion criteria do not specifically mention wanting to quit smoking. How are people being recruited to the study (i.e. what is the message – inviting people to stop smoking, do something about their smoking?) It would also be useful to know the media used to promote participation in this study – posters, email, social media etc.

Page 9, Table 1: In the readiness to quit column, there are only two options (1) ready to quit in the next 30 days or (2) ready to quit in the next 6 months. How were those who were not ready to quit at all considered?

Page 10, line 49: The primary outcome is co-validated 12-month continuous abstinence. Technically, if people quit 4-weeks prior to T0 then this outcome would be measured at 11 months from T0. So, it might be wise to clarify that this 12-month period starts from T0.

How will you handle those who were still smoking at points before T0, but continuously abstinent after T0? I am sure that you have an answer to this, but it is not clear to me in the methodology. I note that participants in the intervention arm who remain continuously abstinent following the joint quit attempt are eligible to receive incentives according to a fixed time schedule'. So, if a participant in the intervention arm is not abstinent a week after the joint quit attempt they is it correct to assume that they will not be entitled to any future incentive? If this is the case, might these people be more likely to drop out?

Page 11, sample size: It is good that data from the CATCH trial, which randomised 604 participants, has been used in the sample size calculations. I have reservations as to whether a tailored approach and the different population will increase the 12-month continuous abstinence rate to the level that you indicate (50%) – I hope that the sample size affords enough power.

One of the key questions for me is 'does a personalised incentive scheme achieve better outcomes than a single scheme'? This, unfortunately, is not something that this study will be able to answer. Given the use of the same treatment, and similar follow-up, in the CATCH study, will there be any ability to match participants between the studies to gain at least some understanding of personalising incentives?

Page 13, line 43: In the readiness to quit analysis, how will you consider those who are not ready to quit at all (these people are technically pre-contemplation, and not contemplating).

One outcome that is missing for me is the additional cost per quitter, compared with usual care. Will this be done? I think that this will be important for policy makers/funders.

REVIEWER	Cheung, Yee Tak Derek The University of Hong Kong
REVIEW RETURNED	03-Mar-2020

GENERAL COMMENTS	1. In the abstract, I suggest the authors to give a bit more details on where and how they recruit participants. Need to state that control group receive no incentive, and all participants joined the group-based programme. Also need to briefly describe the group-based programme. 2. P. 4 Line 10: The authors criticized those NRT and behavioral support have low effectiveness, which contrasts the evidence shown in the reviews cited by the authors. Suggest the authors further describe the context of how these treatments become ineffective. 3. Interventions: Was smoking cessation medication provided to the two trial groups? If so, how much and how long will they obtain? 4. The sequence of the trial procedures is unclear. From what I read I guess: (1) Potential participants join the group-based treatment, (2) receive a letter introducing the trial and incentive, then consent, (3) receive email about group allocation. Its unclear when they complete the baseline questionnaire. I think a recruitment flowchart can clarify this. 5. Do the eligible participants need to measure exhaled CO before consent, so that they are verified as smokers? I think this step is important to avoid gaming. 6. Will the 3-month follow-up be conducted after all the 7 group-based treatment sessions? 7. The authors have not stated the progress of the trial so far.
---

REVIEWER	Marlon Mundt University of Wisconsin Center for Tobacco Research and Intervention, USA
REVIEW RETURNED	10-Apr-2020

GENERAL COMMENTS	This study protocol is for a randomized controlled trial of an incentive-based smoking cessation intervention that provides the intervention subjects a choice between 4 different options of incentive payment schemes. The study is well-planned overall and will add to the literature on incentive-based smoking cessation treatment. My primary concerns with this proposal are: (1) There is likely to be differential loss-to-follow-up in the intervention and control arms. With no incentive to further participate in surveys or in the CO assessments at 3, 6, 9, and 12 months among the control group, participants who are tobacco abstinent may refuse further participation. It would seem reasonable to offer the control group participants some minimal incentive to continue with study assessments, even if there is no abstinence-based contingency to the payment. (2) While this trial will be informative on the choices participants make regarding incentive payment options, there is limited discussion in the article regarding the incentive levels chosen (350-450 Euros) and whether this level of compensation is optimal in terms of cost-effectiveness. Some further discussion of the amount of payments is warranted.
--

VERSION 1 – AUTHOR RESPONSE

Reviewer #1: To the authors:

This protocol describes a study that aims to examine the effect of personalised financial incentives on smoking cessation outcomes, compared to no incentives.

The study aims to provide four different incentive schemes and provides a recommendation on which scheme would be most appropriate depending on a range of different factors. Participants are however free to ignore the recommendation and chose a scheme that they like best. This is a pragmatic approach that lends itself better to implementation in the real world.

Overall this is a well-written protocol paper for a study that is presumably already underway. I have a few comments that might be helpful in clarifying a few points. I have listed these in the order in which they appear in the paper.

1. Page 7, line 36: e-cigarettes are not smoked. I suggest stating 'only using e-cigarettes' instead.

Response: We agree 'only using e-cigarettes' is a better formulation and adjusted this formulation in our manuscript. (p5, paragraph 6)

2. Page 7: It would be helpful to provide a little more information on the group-based treatment programme. It sounds like withdrawal-oriented treatment. Adding some information about the treatment approach, including the use of smoking cessation medication would be useful.

Response: We have added the following information to the group-based smoking cessation training information: (1) The aim of the training is to quit smoking as a group and (2) during the training various types of NRT are discussed. NRT is however not provided through the trial but participants are free to use NRT. See p 6, paragraph 1.

3. I also note that participants stop smoking on the 3rd session, but the that T0 starts at the end of the group programme, which means that some participants would have achieved 4-weeks of abstinence by this time point. It could be helpful to make this clear to readers, as T12, for example, is actually 13 months of abstinence, not 12 months. It might also be useful to have some discussion on the timing of the incentives in relation to the quit date in the group-based treatment programme. Some will have already progressed through four weeks of abstinence, which is when tobacco withdrawal symptoms are most severe. One might argue that incentivising people earlier in the quit attempt could be useful.

Response: Thank you for your suggestions. We have added additional information explaining the one month difference between the abstinence periods and our follow-up time points on page 9, paragraph 2. The incentives are timed such that they coincide with the end of the group training sessions as from that moment onwards participants are 'on their own'; during the group training session the trainer was still available for moments of doubt or issues with withdrawal symptoms. We have added this information on page 6, paragraph 2. We do however agree that further optimising incentives to timing of withdrawal symptoms could be interesting for future research.

4. Page 9, line 8: I am slightly perplexed by the inclusion of readiness-to-quit as a personalising factor when the treatment programme appears to require people to be ready to quit. I suppose the offer of financial incentives may quickly change someone's readiness to quit. I notice that the inclusion/exclusion criteria do not specifically mention wanting to quit smoking. How are people being recruited to the study (i.e. what is the message – inviting people to stop smoking, do something about their smoking?) It would also be useful to know the media used to promote participation in this study – posters, email, social media etc.

Response: We understand the confusion regarding including readiness to quit as a personalising factor. However, even within those ready to quit, there is a difference between those immediately wanting to quit and those considering it in the medium term. We anticipate on the basis of earlier studies that this seemingly subtle difference may make a difference.^{1,2} This is why we used this variable in our algorithm to suggest a suitable incentive scheme.

In our trial participants first sign up to the smoking cessation training, then fill in the questionnaire, and only then are being informed about their allocation to the treatment or control arm. Therefore we assume that readiness to quit is not influenced by the promise of incentives.

Participants are recruited through their employer using several communication outlets, i.e. intranet, email flyers and screensavers. Furthermore information meetings are organised in collaboration with the company responsible for the training, SineFuma. This information is added to page 5, paragraph 7.

5. Page 9, Table 1: In the readiness to quit column, there are only two options (1) ready to quit in the next 30 days or (2) ready to quit in the next 6 months. How were those who were not ready to quit at all considered?

Response: Thank you for this question. As you mention in your previous comment, we anticipate that all participants who sign up for the group-based training are ready to quit at least at some point in the future. That is why we deem the most important difference between those ready to quit within 30 days and those ready to quit within 6 months. Still, you are correct that our algorithm should allow for people who are not ready to quit at all, and so at page 8, paragraph 1 we expanded the (pre-)contemplation stage category now to people who are not ready to quit, and applied this formulation to page 8, table 1 as well.

6. Page 10, line 49: The primary outcome is co-validated 12-month continuous abstinence.

Technically, if people quit 4-weeks prior to T0 then this outcome would be measured at 11 months from T0. So, it might be wise to clarify that this 12-month period starts from T0.

Response: We have added additional information to clarify the abstinence period and its relation to our follow-up time points at page 9, paragraph 2.

7. How will you handle those who were still smoking at points before T0, but continuously abstinent after T0? I am sure that you have an answer to this, but it is not clear to me in the methodology. I note that participants in the intervention arm who remain continuously abstinent following the joint quit attempt are eligible to receive incentives according to a fixed time schedule'. So, if a participant in the intervention arm is not abstinent a week after the joint quit attempt, is it correct to assume that they will not be entitled to any future incentive? If this is the case, might these people be more likely to drop out?

Response: As we want to encourage participants in reaching continuous abstinence, and we do not validate abstinence in this period, we do not punish them for smoking between group session three and our first biochemical validation (i.e. T0). We have clarified this at page 9 paragraph 2.

8. Page 11, sample size: It is good that data from the CATCH trial, which randomised 604 participants, has been used in the sample size calculations. I have reservations as to whether a tailored approach and the different population will increase the 12-month continuous abstinence rate to the level that you indicate (50%) – I hope that the sample size affords enough power.

Response: Thank you for your remark. We assume health care employees are extra motivated as they are extra aware of the health consequences and given the fact that all Dutch health care institutions have to become smoke free by 2025 and the academic hospitals already have to be smoke-free. Although we are aware that a 50% abstinence rate is very high, in relation to the CATCH trial we feel it is reasonable. Also the required sample size in our trial is lower as compared to that of the CATCH trial as randomisation occurs at the individual rather than the company level.

9. One of the key questions for me is 'does a personalised incentive scheme achieve better outcomes than a single scheme'? This, unfortunately, is not something that this study will be able to answer. Given the use of the same treatment, and similar follow-up, in the CATCH study, will there be any ability to match participants between the studies to gain at least some understanding of personalising incentives?

Response: Thank you for your question, this is a highly relevant point. Unfortunately, due to the expected enrolment in our RCT, including a third arm analogous to the CATCH trial to investigate the effectiveness of personalised incentives versus standard incentives was not feasible. Investigators of the CATCH trial are involved as advisors to our project and we are currently exploring what possibilities there are to compare the results of the trials, perhaps through individual meta-analysis techniques for example.

10. Page 13, line 43: In the readiness to quit analysis, how will you consider those who are not ready to quit at all (these people are technically pre-contemplation, and not contemplating).

Response: As we dichotomized our variables, people in the pre-contemplation and the contemplation stage are grouped. We agree that these stages are not the same, but as mentioned at point four above we expect the most important difference to be between preparation and contemplation stage. Furthermore, we do not expect many participants in the pre-contemplation stage as signing up for the group-based training is voluntary.

11. One outcome that is missing for me is the additional cost per quitter, compared with usual care. Will this be done? I think that this will be important for policy makers/funders.

Response: This is indeed very relevant information and we thank you for your valuable suggestion. After the trial is completed we will calculate the Incremental Cost-Effectiveness Ratio (ICER) based on the costs of the intervention and the success rate. Furthermore, the incremental cost utility can be estimated using the EQ-5D. We have added our plans to analyse costs at page 9, paragraph 4.

#Reviewer 2

1. In the abstract, I suggest the authors to give a bit more details on where and how they recruit participants. Need to state that control group receive no incentive, and all participants joined the group-based programme. Also need to briefly describe the group-based programme.

Response: Thank you for your suggestion. In line with your suggestion, the information on recruitment, and the fact that both treated (incentives) and controls (no incentives) are joining the group-based programme, is now added to the abstract.

2. P. 4 Line 10: The authors criticized those NRT and behavioural support have low effectiveness, which contrasts the evidence shown in the reviews cited by the authors. Suggest the authors further describe the context of how these treatments become ineffective.

Response: The reviews cited indeed provide evidence for effectiveness of the interventions. It is however important to note that these effect sizes are relative and not absolute. If in the control group, very few people quit smoking, as is typically the case in cessation trials, the relative effectiveness of the intervention under study may be large but the absolute effectiveness small. That is, in absolute numbers the proportion of smokers that quit is still low. We added this explanation to page 4, paragraph 1.

3. Interventions: Was smoking cessation medication provided to the two trial groups? If so, how much and how long will they obtain?

Response: Smoking cessation medication was not provided as part of the trial, however, during the group based training sessions the potential usefulness of various types of medication was discussed. Participants were free to use medication as part of their quit attempt. In the Netherlands since 1st January 2020 nicotine replacement therapy and cessation medication are available through health insurance without using one's deductible.

4. The sequence of the trial procedures is unclear. From what I read I guess: (1) Potential participants join the group-based treatment, (2) receive a letter introducing the trial and incentive, then consent, (3) receive email about group allocation. It's unclear when they complete the baseline questionnaire. I think a recruitment flowchart can clarify this.

Response: Thank you for your suggestion. We have adjusted the location of our flowchart to page 6 to ensure readers understand the process.

5. Do the eligible participants need to measure exhaled CO before consent, so that they are verified as smokers? I think this step is important to avoid gaming.

Response: CO is measured at the start of the group-based training by the group trainer, participants provide consent to include this measure in our dataset. A previous study using incentives for smoking cessation among pregnant women showed there was little evidence for gaming.³ Furthermore, the CATCH trial reported no reasons to suspect gaming, likely due to the need to follow 7 group-based training sessions as part of the trial procedure.⁴ Therefore, we decided not to use the CO-measurement as a prerequisite for inclusion in our trial.

6. Will the 3-month follow-up be conducted after all the 7 group-based treatment sessions?

Response: All group-based training sessions will be completed before the first CO measurement in our trial, T=0. We have added additional information to clarify the difference between our follow-up time points and the total abstinence period, as also mentioned by reviewer 1. See page 9, paragraph 2.

7. The authors have not stated the progress of the trial so far.

Response: Thank you for this remark, we have added the status of our trial in the protocol. See page 14, paragraph 1.

reviewer 3 to the authors:

This study protocol is for a randomized controlled trial of an incentive-based smoking cessation intervention that provides the intervention subjects a choice between 4 different options of incentive payment schemes. The study is well-planned overall and will add to the literature on incentive-based smoking cessation treatment. My primary concerns with this proposal are:

1. There is likely to be differential loss-to-follow-up in the intervention and control arms. With no incentive to further participate in surveys or in the CO assessments at 3, 6, 9, and 12 months among the control group, participants who are tobacco abstinent may refuse further participation. It would seem reasonable to offer the control group participants some minimal incentive to continue with study assessments, even if there is no abstinence-based contingency to the payment.

Response: Thank you for your comment. Loss to follow up among the control group is a valid concern. However, results from the CATCH trial showed only 4% was lost to follow-up in the control group and 3% withdrew in a 12 month period.⁴ Furthermore, participants receive the training from their employer and given the duration of their contracts typically remain employed during follow-up, which makes it relatively easy to trace them and might make them more motivated to keep participating.

2. While this trial will be informative on the choices participants make regarding incentive payment options, there is limited discussion in the article regarding the incentive levels chosen (350-450 Euros) and whether this level of compensation is optimal in terms of cost-effectiveness. Some further discussion of the amount of payments is warranted.

Response: Thank you for your comment. We agree that this information was lacking. The size of incentives should be large enough to motivate smoking cessation, but not too large either to avoid crowding out intrinsic motivation and to make scaling up the trial prohibitively costly.⁵ The standard scheme (€350,-) is identical to the scheme in the CATCH trial.⁴ Given time discounting due to inflation and individual impatience, it seems natural to apply some compensation in the total potential gains to the descending and ascending scheme. For example, why would someone self-select into an ascending scheme with zero pay-out in the first six months if the total potential gain is the same as the descending scheme where one can cash-in €150,- already after at baseline? Further, as deposit contracts have proven to be relatively most effective among those who choose one,² we aimed to make this scheme the most attractive by providing the largest total reward size. This information is added to page 7, paragraph 4.

We thank you for the opportunity to let us improve the quality of our work and thank the peer-reviewers for their constructive feedback. We trust that the revisions are to your satisfaction, however, feel free to come back to us if any further information or clarification is needed. To the editor we would like to note that we exceeded the 4000 words.

With kind regards,

Nienke W Boderie, Hans van Kippersluis, Diarmaid T Ó Ceallaigh, Márta K Rado, Alex Burdorf, Frank J van Lenthe and Jasper V Been

1. Volpp KG, Troxel AB, Pauly MV, et al. A randomized, controlled trial of financial incentives for smoking cessation. *N Engl J Med* 2009;360(7):699-709.

2. Halpern SD, French B, Small DS, et al. Randomized trial of four financial-incentive programs for smoking cessation. *N Engl J Med* 2015;372(22):2108-17.

3. Ierfino D, Mantzari E, Hirst J, et al. Financial incentives for smoking cessation in pregnancy: a single-arm intervention study assessing cessation and gaming. *Addiction* 2015;110(4):680-88.
4. van den Brand FA, Nagelhout GE, Winkens B, et al. Effect of a workplace-based group training programme combined with financial incentives on smoking cessation: a cluster-randomised controlled trial. *The Lancet Public Health* 2018;3(11):e536-e44.
5. Gneezy U, Meier S, Rey-Biel P. When and why incentives (don't) work to modify behavior. *J Econ Perspect* 2011;25(4):191-210.

VERSION 2 – REVIEW

REVIEWER	Hayden McRobbie NDARC University of New South Wales Australia
REVIEW RETURNED	05-Jun-2020

GENERAL COMMENTS	Thank you for asking me to review the revision of this paper. The authors have addressed my comments in their revision. There is greater clarification of the measurement time points and readiness to quit. It's good that the authors have also added cost-effectiveness measures.
---

REVIEWER	Cheung, Yee Tak Derek The University of Hong Kong, Hong Kong
REVIEW RETURNED	05-Jun-2020

GENERAL COMMENTS	The authors have fully addressed my comments.
---